# Enhanced Mechanical Properties of Multiscale Carbon Fiber/Epoxy Unidirectional Composites with Different Dimensional Carbon Nanofillers

**DOI:** 10.3390/nano10091670

**Published:** 2020-08-26

**Authors:** Yu Liu, Dong-Dong Zhang, Guang-Yuan Cui, Rui-Ying Luo, Dong-Lin Zhao

**Affiliations:** 1Research Institute for Frontier Science, Beihang University, Beijing 100029, China; liuyu9175@buaa.edu.cn (Y.L.); m15300178689_3@163.com (G.-Y.C.); 2State Key Laboratory of Chemical Resource Engineering, Key Laboratory of Carbon Fiber and Functional Polymers, Ministry of Education, Beijing University of Chemical Technology, Beijing 100029, China; zhangdd8888@126.com

**Keywords:** graphene, carbon nanotubes, epoxy, mechanical property, composites

## Abstract

Ammonia modified graphene-carbon nanotubes/continuous carbon fiber reinforced epoxy unidirectional multiscale composites (AMGNS-MWCNT/CFEP) were prepared by adding ammonia modified graphene and carbon nanotubes to an epoxy matrix to reduce agglomeration of carbon nanofillers in the epoxy matrix and improve composites properties. Fourier transform infrared spectroscopy (FTIR), scanning electron microscope (SEM), and universal testing machines were used to characterize the properties of carbon nanofillers, AMGNS-MWCNT/epoxy nanocomposites, and AMGNS-MWCNT/CFEP unidirectional composites. When the AMGNS-MWCNT content was 1.0 wt%, flexural strength, the flexural modulus and interlaminar shear strength of AMGNS-MWCNT/CFEP unidirectional composites reached the maximum value of 1520.3 MPa, 138.88 GPa, and 87.80 MPa, respectively, which were 12.5%, 9.42%, and 10.1% higher than that of carbon fiber reinforced epoxy unidirectional composites (CFEP). The synergistic mechanism of two carbon nanofillers in the matrix is discussed.

## 1. Introduction

Carbon fiber reinforced polymers (CFRPs) have excellent mechanical properties, low density, specific stiffness, and strength, and have important applications in aerospace, automobile, sports goods, and other industries [1,2,3,4]. Epoxy is a kind of thermosetting resin, which is widely used as the matrix of CFRPs. However, the inherent brittleness of epoxy limits its applications. In practical applications, nanomaterials are the most promising strengthening and toughening reinforcements for carbon fiber/epoxy composites. According to the different constraints in the space dimension, nanomaterials can be divided into zero-dimensional, one-dimensional, and two-dimensional nanomaterials. The nanomaterials in different dimensions have anisotropy to the load transfer direction [5]. Nanomaterials modified epoxy composites generally contain only one kind of nanomaterial of a single dimension [6,7,8]. However, the properties of the epoxy composites reinforced by single dimensional nanofillers will be improved while causing other properties to degrade [9,10,11,12]. To solve this problem, two carbon nanofillers with different dimensions are designed to be added to the epoxy matrix to improve the comprehensive performance of CFRPs [13,14].

The surface and edge of graphene oxide prepared by the Hummers method contain a large number of oxygen-containing functional groups [15,16,17]. The oxygen-containing functional groups provide the possibility of introducing modified substances on the surface of graphene. Therefore, modification of graphene can be achieved by modifying and reducing the graphene oxide [18,19]. Graphene prepared by the graphene oxide reduction method, which is widely used in the laboratory, often has certain structural defects, and the carbon atoms located in defects have strong chemical activity, so it is also possible to modify graphene at this active site. The covalent modification of graphene mainly involves four reactions: nucleophilic substitution, electrophilic addition, condensation, and addition polymerization [20,21,22,23,24,25,26,27,28,29]. The lone electron pair in ammonia reacts with the oxygen-containing groups on the surface and edges of graphene, mainly by nucleophilic substitution. One is to produce Brønsted acid by NH_4_^+^, and the other is to produce Lewis acid in the form of -NH_2_ [30].

In this study, ammonia modified graphene and carbon nanotubes were prepared and used as reinforcements to form carbon fiber/epoxy unidirectional multiscale composites. The combination of graphene nanosheets and carbon nanotubes can effectively prevent the agglomeration and winding by taking advantage of their differences in the direction of force transfer in spatial dimensions. The mechanical properties of the composites were improved, and the synergistic mechanism of ammonia modified graphene and carbon nanotubes was studied.

## 2. Materials and Methods

### 2.1. Preparation of Ammonia Modified Graphene-Carbon Nanotubes/Continuous Carbon Fiber Reinforced Epoxy Composites

Graphene nanosheets (GNS) were prepared by the chemical oxidation-rapid thermal expansion reduction-ultrasonic stripping method [31]. A total of 0.5 g graphite oxide was quickly put into a preheated muffle furnace at 1050 °C and kept for 30 s to obtain expanded graphite. After sieving, the expanded graphite was dissolved into anhydrous ethanol at a ratio of 0.2 g/100 mL, and ultrasonic treatment was performed for 15 h. After the ethanol was volatilized, the sample was dried in a vacuum oven at 120 °C for 4 h to obtain GNS. Ammonia modified graphene nanosheets (AMGNS) were prepared by mixing 90 mL hydrogen peroxide and 150 mL ammonia with 1.5 g GNS. In the preparation process, the mixture was ultrasonically treated for 4 h. After filtration, the filter cake was washed to neutral with deionized water, and then dried at 65 °C inside the vacuum oven for 12 h.

A certain amount of carbon nanotubes (CNT) were refluxed with 100 mL concentrated nitric acid for 5 h. After filtration, the filter cake was washed to neutral with deionized water, and then dried at 60 °C inside the vacuum oven for 24 h.

Equally proportional AMGNS and modified multi-wall carbon nanotubes (MWCNT) (0.1 wt%, 0.5 wt%, 1.0 wt%, 2.0 wt%, and 3.0 wt%) were added into the resin solution. The mass ratio of epoxy (E-51, Sinopec Baling Petrochemical Co., Hunan, China) to curing agent (MeTHPA, Beijing Chemical Reagent Co., Beijing, China) was 10:7. After mechanical stirring and ultrasonic reaction for 6 h, a small amount of promoters (DMP-30, Beijing Chemical Reagent Co., Beijing, China) were added, and the ultrasonic reaction continued for 0.5 h. Then, the reaction solution was placed in a vacuum oven at 50 °C to debubble for 0.5 h. The mixed solution was poured into a stainless-steel mold and cured in sections with temperature rise. AMGNS-MWCNT/epoxy nanocomposites were obtained after demolding. The neat epoxy and AMGNS-MWCNT/epoxy nanocomposites of different components are named as EP, CG0.1, CG0.5, CG1.0, CG2.0, and CG3.0.

The process was repeated and the mixture was poured into a stainless-steel mold with a copper wire frame wound with T300 (Toray Industries Co., Tokyo, Japan, 3K) continuous carbon fiber. AMGNS-MWCNT/continuous carbon fiber reinforced epoxy unidirectional composites (AMGNS-MWCNT/CFEP) was obtained by curing, stripping, and processing procedures. The volume fraction of carbon fiber was 60 vol%. The contents of AMGNS-MWCNT were 0 wt%, 0.1 wt%, 0.5 wt%, 1.0 wt%, and 2.0 wt% with an equally proportion. The neat continuous carbon fiber reinforced epoxy unidirectional composite and AMGNS-MWCNT/CFEP unidirectional composites of different components are named as CFEP, CGFP0.1, CGFP0.5, CGFP1.0, and CGFP2.0.

### 2.2. Characterization

The surface functional groups of AMGNS-MWCNT were investigated by Fourier transform infrared spectroscopy (FT-IR, Nicolet, Nexus670, Madison, WI, USA). The surface morphology of composites was investigated by scanning electron microscope (SEM, Hitachi, S-4700, Tokyo, Japan). The tensile, flexural, and interlaminar shear properties of composites were tested by universal testing machines (Instron, Instron^TM^ 1185 and Instron^TM^ 5567, Instron Corporation, Canton, MA, USA).

The tensile strength was calculated using Equation (1):(1)$$\sigma_{t}=\frac{P}{bh}$$
where *σ_t_* is the tensile strength (MPa); *P* is the maximum load (N); *b* is the specimen width (mm); and *h* is the specimen thickness (mm).

The tensile modulus was calculated using Equation (2):(2)$$E_{t}=\frac{L_{0}∆P}{bh∆L}$$
where *E_t_* is the tensile modulus (MPa); *L*_0_ is the gauge length (mm); Δ*P* is the load increment (N); Δ*L* is the gauge length increment corresponding to Δ*P* (mm).

Elongation at break was calculated using Equation (3):(3)$$\varepsilon_{t}=\frac{∆L_{b}}{L_{0}}\times 100\%$$
where *ε_t_* is the elongation at break (%); and Δ*L_b_* is the elongation at break within the gauge length *L*_0_ of the sample (mm).

The flexural strength was calculated using Equation (4):(4)$$\sigma_{f}=\frac{3PL}{2bh^{2}}$$
where *σ_f_* is the flexural strength (MPa) and *L* is the span (mm).

The flexural strain was calculated using Equation (5):(5)$$\varepsilon_{f}=\frac{6sh}{L^{2}}\times 100\%$$
where *ε_f_* is the flexural strain (%) and *s* is the deflection (mm).

The flexural modulus was calculated using Equation (6):(6)$$E_{f}=\frac{L^{3}∆P}{4bh^{3}∆S}$$
where *E_f_* is the flexural modulus (MPa) and Δ*S* is the deflection increment corresponding to Δ*P* (mm).

The interlaminar shear strength was calculated using Equation (7):(7)$$\tau_{s}=\frac{3P_{b}}{4bh}$$
where *τ_s_* is the interlaminar shear strength (MPa) and *P_b_* is the maximum load (N).

## 3. Results and Discussion

### 3.1. The Properties of AMGNS and Modified CNT

To improve the bonding performance of GNS and CNT to the epoxy matrix, the surface modification of these two carbon nanofillers was carried out. The edge of GNS prepared by the thermal reduction of graphite oxide contained many carboxyl functional groups. These active groups can react with ammonia to obtain AMGNS. The FTIR spectrum of GNS before and after ammonia modification are shown in Figure 1a. In the ammonia modified GNS spectrum, the peaks appeared at 1560 cm^−1^ (–NH_2_ stretching), 1400 cm^−1^ (C–N stretching) [32,33], and blue shift (shift to lower wavenumbers) of all characteristic peaks, indicating that ammonia had successfully modified graphene. FTIR spectrum of CNT treated by concentrated nitric acid refluxes are shown in Figure 1b. The peaks appearing at 1732 cm^−1^ (C=O stretching) indicate that the surface of the CNT had been modified by oxygen-containing functional groups [1].

### 3.2. Mechanical Properties of AMGNS-MWCNT/Epoxy Nanocomposites

The tensile strength, tensile modulus, and elongation at break of AMGNS-MWCNT/epoxy nanocomposites are shown in Table 1 and Figure 2. With the increase in the content of carbon nanofillers, the tensile strength and elongation at break of the composites increased first and then decreased. When the content of AMGNS-MWCNT was 2.0 wt%, the tensile strength and elongation at break of the AMGNS-MWCNT/epoxy nanocomposites reached the maximum value of 63.53 MPa and 3.53%, respectively. Compared with neat epoxy, the tensile strength and elongation at break of AMGNS-MWCNT/epoxy nanocomposites increased by 20.7% and 34.7%, respectively. When the content of AMGNS-MWCNT was 0.1 wt%, the tensile strength and elongation at break of AMGNS-MWCNT/epoxy nanocomposites were not significantly different from that of neat epoxy. This is because the content of AMGNS-MWCNT in the resin matrix was too low, there was no good contact between the nanofillers, and they were not evenly dispersed in the matrix, which tended to form defects in the materials. Therefore, the nanocomposites are more likely to fracture when subjected to external forces. When the content of nanofillers increased to a certain extent (3 wt%), the agglomeration of AMGNS or MWCNT became more serious, and their dispersion in the epoxy matrix became worse, resulting in more internal defects, which reflected poor mechanical properties. The tensile modulus of the nanocomposites increased with the content of AMGNS-MWCNT. When the content of AMGNS-MWCNT was 3.0 wt%, the tensile modulus of the AMGNS-MWCNT/epoxy nanocomposites reached the maximum value of 3.07 GPa, which was 24.8% higher than that of neat epoxy.

SEM images of the fractured surface of AMGNS-MWCNT/epoxy nanocomposites are shown in Figure 3. When the content of AMGNS-MWCNT was low (<0.5 wt%), the fractured surface was relatively smooth, and MWCNT was sporadically distributed in the epoxy matrix, which makes it difficult to strengthen the epoxy resin. When the content of AMGNS-MWCNT was high (>3.0 wt%), the agglomeration of MWCNT was very serious, and the GNS did not have a good coupling effect with CNT, which would easily lead to the deterioration of the mechanical properties of composites. When the content of AMGNS-MWCNT was within a certain range (1.0–2.0 wt%), AMGNS-MWCNT were relatively dispersed in the matrix, and the agglomeration and entangling phenomenon were not obvious. This is because one-dimensional long-range ordered CNT can bridge the GNS, improve the specific surface area of the AMGNS-MWCNT filler, and reduce the degree of agglomeration, thus contributing to the improvement of the mechanical properties of the composite.

The flexural strength, flexural modulus, and flexural strain of the AMGNS-MWCNT/epoxy nanocomposites are shown in Table 2 and Figure 4. With the increase in AMGNS-MWCNT content, the flexural strength, flexural modulus, and flexural strain of AMGNS-MWCNT/epoxy nanocomposites increased first and then decreased. When the content of AMGNS-MWCNT was 2.0 wt%, the flexural strength and flexural strain of the AMGNS-MWCNT/epoxy nanocomposites reached a maximum of 108.47 MPa and 3.23%, respectively. Compared with neat epoxy, the flexural strength and flexural strain of the AMGNS-MWCNT/epoxy nanocomposites increased by 55.5% and 11.8%, respectively. When the content of AMGNS-MWCNT was 1.0 wt%, the flexural modulus of the AMGNS-MWCNT/epoxy nanocomposites reached a maximum of 3.99 GPa. Compared with neat epoxy, the flexural modulus increased by 23.5%.

### 3.3. Mechanical Properties of AMGNS-MWCNT/CFEP Unidirectional Composite

The flexural properties and interlaminar shear strength for AMGNS-MWCNT/CFEP unidirectional composites are shown in Table 3. The flexural strength and flexural modulus of AMGNS-MWCNT/CFEP unidirectional composite with different AMGNS-MWCNT contents are shown in Figure 5. With the increase in the AMGNS-MWCNT content, the flexural strength and flexural modulus of the AMGNS-MWCNT/CFEP unidirectional composite increased first and then decreased. When the content of AMGNS-MWCNT was 1.0 wt%, the flexural strength and flexural modulus of the AMGNS-MWCNT/CFEP unidirectional composite reached the maximum value of 1520.3 MPa and 138.88 GPa, respectively. Compared with the CFEP composite without any fillers, the flexural strength and flexural modulus increased by 12.5% and 9.42%, respectively.

SEM images of the flexural fracture surface of the AMGNS-MWCNT/CFEP unidirectional composites with a content of 1.0 wt% of AMGNS-MWCNT are shown in Figure 6. The volume fraction of carbon fiber was 60 vol%. The carbon fiber was evenly distributed in the composite, and there was no obvious crack around it. It can be seen from Figure 6b that when the AMGNS-MWCNT/CFEP unidirectional composite was broken, the pull-out effect of carbon fibers was dominant. This is because the addition of one-dimensional CNT can prevent the agglomeration of GNS and reduce the formation of internal defects in the material. The AMGNS-MWCNT fillers form a good bridging effect between the carbon fibers and epoxy matrix. When an external force is applied, the synergy between the two nanofillers can effectively delay the crack propagation to carbon fibers, and the extraction of carbon fibers and the interaction of fillers consume a large amount of applied energy, which makes the composite have excellent mechanical properties.

Interlaminar shear strength of the AMGNS-MWCNT/CFEP unidirectional composites with different AMGNS-MWCNT contents are shown in Figure 7. With the increase in AMGNS-MWCNT content, the interlaminar shear strength of AMGNS-MWCNT/CFEP unidirectional composites increased first and then decreased. When the content of AMGNS-MWCNT was 1.0 wt%, the interlaminar shear strength of the AMGNS-MWCNT/CFEP unidirectional composite reached the maximum value of 87.80 MPa, which was 10.1% higher than that of the CFEP without any fillers. This indicates that the AMGNS-MWCNT/CFEP unidirectional composites had good interlaminar interfacial properties and the bonds between the fibers were tight. Compared with the traditional carbon fiber composite materials, the existence of the two-dimensional nanofillers not only exerts the synergistic effect between the fillers, but also has the dimensional effect between the two fillers. Due to the different directions of conduction when different dimensional materials face the stress, the AMGNS-MWCNT/CFEP unidirectional composites can rapidly and effectively transfer the stress to the three-dimensional direction under the action of the applied load. When these stresses meet the GNS and CNT, the formation of cracks will be slowed down. Therefore, the AMGNS-MWCNT/CFEP unidirectional composites have relatively high interlaminar shear strength.

The dispersion and compatibility of two-dimensional GNS and one-dimensional CNT in the epoxy matrix play an important role in enhancing the mechanical properties of composites. Figure 8 shows the simulation of the strengthening and toughening mechanism of two-dimensional GNS and one-dimensional CNT fillers in composites. Due to its large length-diameter ratio, CNTs are easy to be wound and agglomerated in the matrix, which limits their excellent performance. GNS are also prone to pile-up in the matrix due to their van der Waals’ forces and π–π bonding, which results in their inability to give full play to its performance as a load transfer in the composite material. When both fillers are used as reinforcing materials for the composites, the defects are less likely to form in the case that the two fillers are well dispersed. The fibrous CNTs effectively limit the agglomeration between GNS due to the π–π bond. GNS and CNTs are combined to form a three-dimensional network structure, which improves the interface contact area between the reinforcement and the matrix and generates a strong molecular binding force. When subjected to external forces, the load can develop along the three-dimensional direction, thus improving the fracture resistance of the composite.

## 4. Conclusions

This study demonstrates the improvement in the mechanical properties of multiscale carbon fiber/epoxy unidirectional composites with different dimensional carbon nanofillers. When the content of AMGNS-MWCNT was 2.0 wt% and 1.0 wt%, the AMGNS-MWCNT/epoxy nanocomposites and AMGNS-MWCNT/CFEP unidirectional composites showed the best performance, respectively. The tensile strength, elongation at break, flexural strength, and flexural strain of the AMGNS-MWCNT/epoxy nanocomposites could reach to 63.53 MPa, 3.53%, 108.47 MPa, and 3.23%, respectively, which were 20.7%, 34.7%, 55.5%, and 11.8% higher than that of neat epoxy. Furthermore, the flexural strength, flexural modulus, and interlaminar shear strength of the AMGNS-MWCNT/CFEP unidirectional composites reached the maximum value of 1520.3 MPa, 138.88 GPa, and 87.80 MPa, respectively, which were 12.5%, 9.42%, and 10.1% higher than that of CFEP. This is because two-dimensional GNS and one-dimensional CNT fillers are combined to form a three-dimensional network structure in composites to improve the interface contact area between the reinforcement and the matrix. The three-dimensional network structure can effectively disperse the load and improve the mechanical properties of the composites.

## Figures and Tables

**Figure 1 nanomaterials-10-01670-f001:**
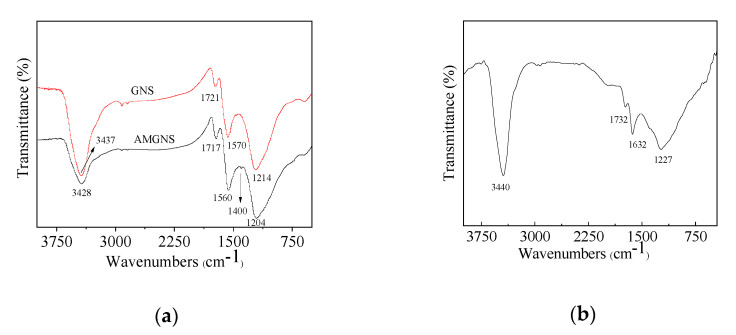
FTIR spectrum of (**a**) GNS and AMGNS; (**b**) CNT after concentrated nitric acid treatment.

**Figure 2 nanomaterials-10-01670-f002:**
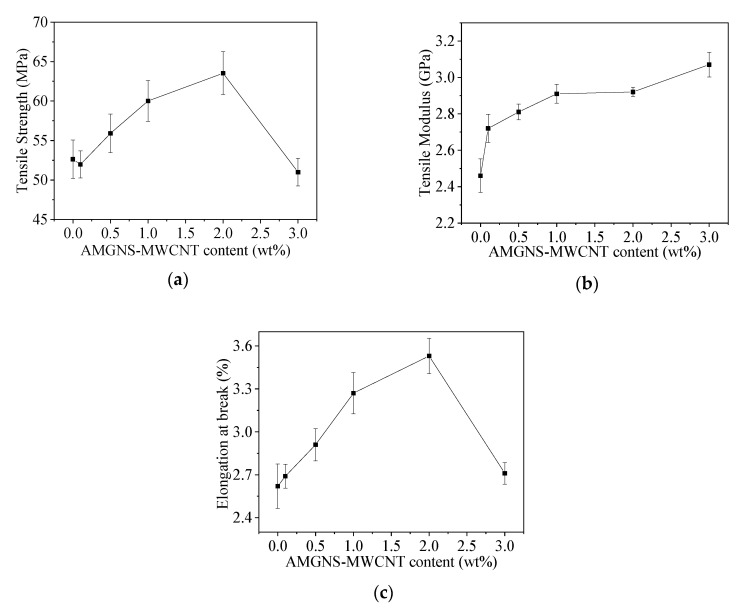
Tensile properties of the AMGNS-MWCNT/epoxy nanocomposites: (**a**) tensile strength; (**b**) tensile modulus; (**c**) elongation at break.

**Figure 3 nanomaterials-10-01670-f003:**
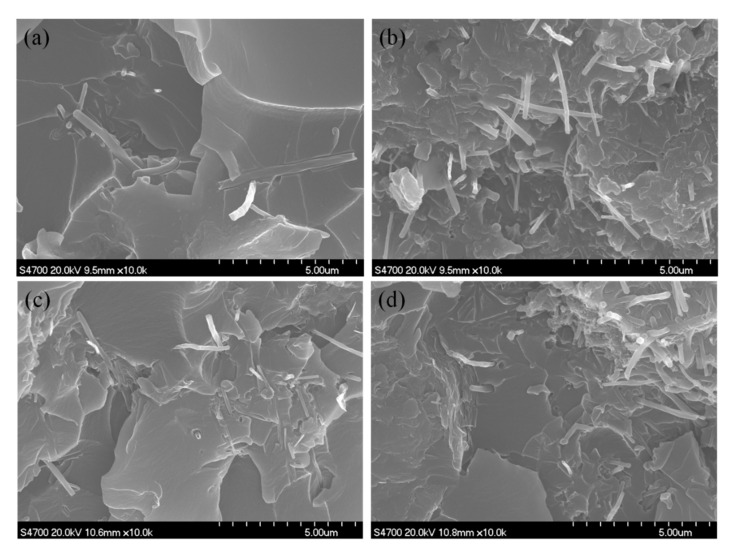
SEM micrographs of the fractured surface of the epoxy composites with AMGNS-MWCNT mixture contents of (**a**) 0.5 wt%, (**b**) 1.0 wt%, (**c**) 2.0 wt%, and (**d**) 3.0 wt%. (Appendix A).

**Figure 4 nanomaterials-10-01670-f004:**
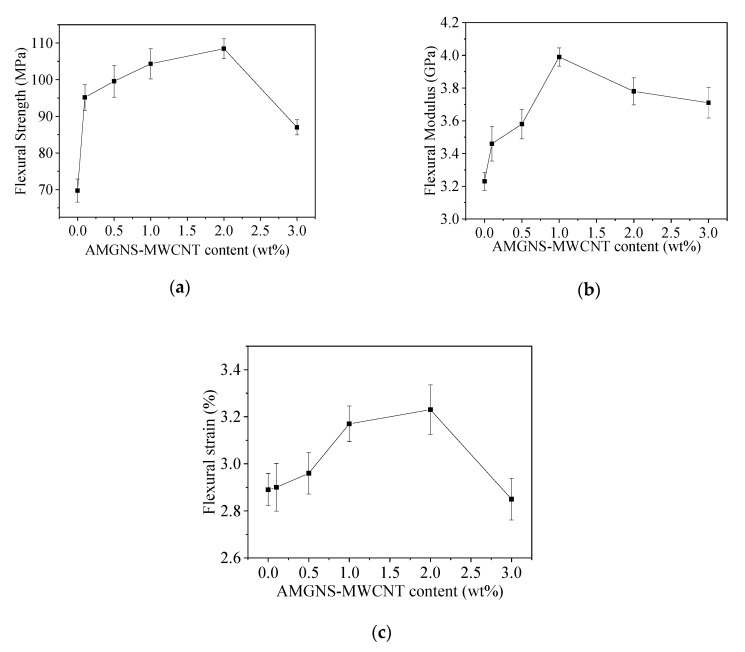
Flexural properties of the AMGNS-MWCNT/epoxy nanocomposites: (**a**) flexural strength; (**b**) flexural modulus; (**c**) flexural strain.

**Figure 5 nanomaterials-10-01670-f005:**
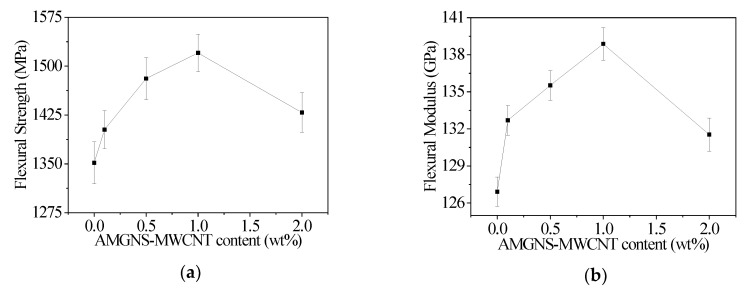
Flexural properties of the AMGNS-MWCNT/CFEP unidirectional composites: (**a**) flexural strength; (**b**) flexural modulus.

**Figure 6 nanomaterials-10-01670-f006:**
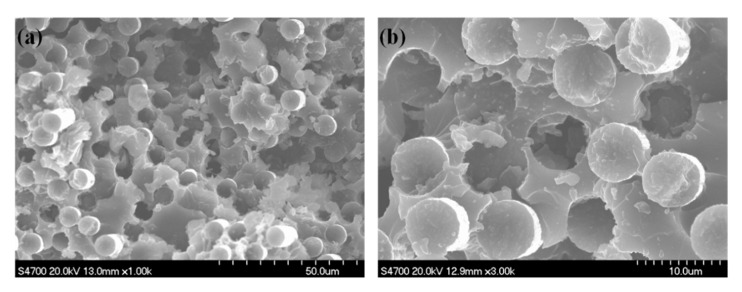
SEM images of the AMGNS-MWCNT/CFEP unidirectional composites: (**a**) low-magnification micrograph of the fiber and surrounding matrix; (**b**) high-magnification view of the fiber/matrix interface.

**Figure 7 nanomaterials-10-01670-f007:**
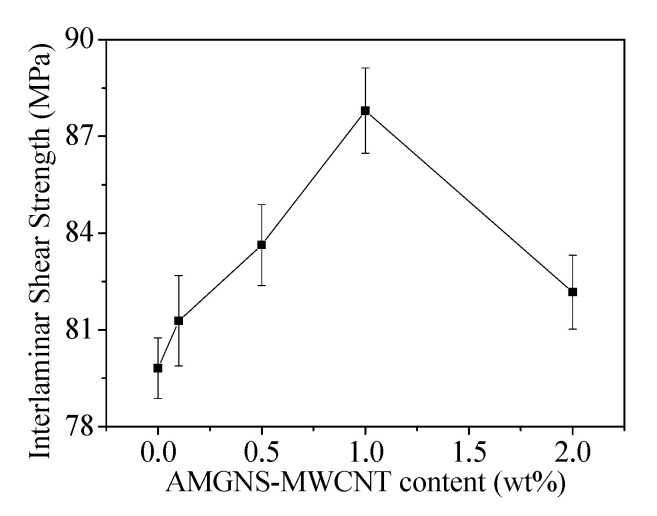
Interlaminar shear strength of the AMGNS-MWCNT/CFEP unidirectional composites.

**Figure 8 nanomaterials-10-01670-f008:**
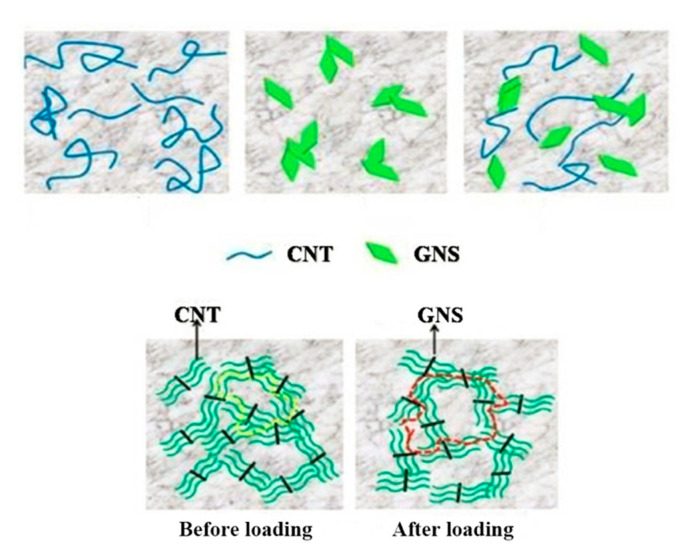
Schematic images of the reinforcement dispersion in the epoxy composites.

**Table 1 nanomaterials-10-01670-t001:** Tensile properties of AMGNS-MWCNT/epoxy nanocomposites with different AMGNS-MWCNT content.

Specimens	Tensile Strength/MPa	Tensile Modulus/GPa	Elongation at Break/%
EP	52.63	2.46	2.62
CG0.1	51.97	2.72	2.69
CG0.5	55.91	2.81	2.91
CG1.0	60.02	2.91	3.27
CG2.0	63.53	2.92	3.53
CG3.0	50.99	3.07	2.71

**Table 2 nanomaterials-10-01670-t002:** Flexural properties of the AMGNS-MWCNT/epoxy nanocomposites with different AMGNS-MWCNT content.

Specimens	Flexural Strength/MPa	Flexural Modulus/GPa	Flexural Strain/%
EP	69.75	3.23	2.89
CG0.1	95.19	3.46	2.90
CG0.5	99.58	3.58	2.96
CG1.0	104.34	3.99	3.17
CG2.0	108.47	3.78	3.23
CG3.0	87.01	3.71	2.85

**Table 3 nanomaterials-10-01670-t003:** Flexural properties and interlaminar shear strength of the AMGNS-MWCNT/CFEP unidirectional composites with different AMGNS-MWCNT content.

Specimens	Flexural Strength/MPa	Flexural Modulus/GPa	Interlaminar Shear Strength/MPa
CFEP	1351.7	126.92	79.81
CGFP0.1	1402.6	132.69	81.28
CGFP0.5	1480.9	135.52	83.63
CGFP1.0	1520.3	138.88	87.80
CGFP2.0	1428.8	131.54	82.17

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
