# Peer review of "Enhanced Mechanical Properties of Multiscale Carbon Fiber/Epoxy Unidirectional Composites with Different Dimensional Carbon Nanofillers"

_nanomaterials, 2020, doi:10.3390/nano10091670_

Round 1

Reviewer 1 Report

The authors in the paper showed the synergistic mechanism of two carbon nanofillers namely ammonia modified Graphene nanosheet and Carbon nanotube mixed with epoxy in improving the mechanical behavior of composites. They carried out the research systematically by changing the content of composites and measuring the mechanical strength of each composites. The results presented in the paper were discussed logically with enough evidences and references. With some minor grammatical corrections, I believe this paper can be accepted in the present format for the publication. 

Reviewer 2 Report

The manuscript by Liu et al. deals with the enhancement of the mechanical properties of a carbon reinforced epoxy composite by using various carbon nanomaterials as modified graphene and carbon nanotubes apart from carbon fibers. It could be a publishable article upon major revision. I have the following comments:

Lines 62-63, please explain the “chemical oxidation-rapid thermal expansion reduction-ultrasonic stripping method” and add either details or citation (with short description) if procedure is elsewhere reported in detail.

FT-IR discussion needs extensive revision. Authors don’t seem to have the proper background for peak assignment and explanation. For example, flexural is used for beams and not IR (bending) vibration modes, you assign both 1400 and 727 cm-1 peaks as C-N stretching mode, which is impossible, while you assign peaks generally as vibration modes. You should read and cite liable bibliography regarding IR.

Claim in lines 130-131 cannot be supported or concluded by FT-IR. It should be deleted.

You should follow the same principles for presenting same similar graphs. I.e. figure 1a and 1b.

Figure 2c and Figure 3. Isn’t elongation at break the same as fracture strain?

Lines 164-167, what do you mean and why? You mention agglomeration in line 147, but this statement is confusing. Moreover, I can see a spot on top right of Figure 4d where you do have agglomerated CNTs but is this the case for the whole material? Maybe you should present relevant spots for the 3% sample as SI.

You should add a table including mechanical properties data for all samples so as comparison for readers is easier.

You should rewrite conclusions section since this is just results repetition. It is not conclusions.

Manuscript needs extensive English editing, for example lines 63, 66, 69, 80, 136, GO in line 41 and many more.

Round 2

Reviewer 2 Report

Authors have made appropriate modifications in the manuscript. Basically, they have altered FT-IR, deleted Figure 3 and added further SEM images supporting their results. Although, it is strange and disappointing that they hadn't noticed the vital differences between Figure 2c and 3 and it is obvious that they don't know FT-IR, this is a publishable work. I recommend minor revision just for text editing by experts, no need for further reviewing process.

Author Response

The texts of FT-IR part were further revised. The explanation of blue shift in line 130 was added as “shift to lower wavenumbers”.

Incorrect writings in manuscript were corrected. The corrections are as follows.

“adding ammonia modified graphene and carbon nanotubes to the epoxy matrix”  in line 14 and 15 was corrected as “adding ammonia modified graphene and carbon nanotubes to epoxy matrix”.

“universal testing machine” in line 17 was corrected as “universal testing machines”.

“According to their different constraints in the space dimension” in line 33 was corrected as “According to the different constraints in the space dimension”.

“unidimensional nano-enhanced epoxy composites will improve some properties while causing other properties to degrade” in line 37 and 38 was corrected as “the properties of epoxy composites reinforced by single dimensional nanofillers will be improved while causing other properties to degrade”.

Indenting in line 41 was added.

“The presence of these oxygen-containing functional groups provides the possibility of introducing modified substances on the surface of graphene”  in line 42 and 43 was corrected as “The oxygen-containing functional groups provide the possibility of introducing modified substances on the surface of graphene”.

“and the carbon atoms located in the defects have strong chemical activity” in line 46 was corrected as “and the carbon atoms located in defects have strong chemical activity”.

“1.5g” in line 67 was corrected as “1.5 g”.

“4h” in line 68 was corrected as “4 h”.

“A certain amount of carbon nanotubes (CNT) was refluxed with 100 mL concentrated nitric acid for 5 h.” in line 70 and 71 was corrected as “A certain amount of carbon nanotubes (CNT) were refluxed with 100 mL concentrated nitric acid for 5 h.”

“a small amount of promoter (DMP-30, Beijing Chemical Reagent Co.) was added” in line 76 and 77 was corrected as “a small amount of promoters (DMP-30, Beijing Chemical Reagent Co.) were added”.

“The contents of AMGNS-MWCNT were 0 wt%, 0.1 wt%, 0.5 wt%, 1.0 wt%, and 2.0 wt% with an equally proportional” in line 87 and 88 was corrected as “The contents of AMGNS-MWCNT were 0 wt%, 0.1 wt%, 0.5 wt%, 1.0 wt%, and 2.0 wt% with an equally proportion”.

“a universal testing machine” in line 96 was corrected as “universal testing machines”.

“there was no good contact between them” in line 146 and 147 was corrected as “there was no good contact between the nanofillers”.

“which tended to form defects in the materials” in line 147 and 148 was corrected as “which tended to form defects in the materials”.

“nanocomposite” in line 154 was corrected as “nanocomposites”.

“the agglomeration of MWCNT were very serious” in line 165 was corrected as “the agglomeration of MWCNT was very serious”.

“the AMGNS-MWCNT content” in line 180 was corrected as “the content of AMGNS-MWCNT”.

“the AMGNS-MWCNT content” in line 184 was corrected as “the content of AMGNS-MWCNT”.

“ILSS” in line 193 was corrected as “Interlaminar shear strength”.

Space character in the middle of line 194 was added.

“without any filler” in line 200 was corrected as “without any fillers”.

“ILSS” in line 202 was corrected as “Interlaminar shear strength”.

“The carbon fiber was evenly distributed in the composite” in line 209 was corrected as “The carbon fibers were evenly distributed in the composite”.

“the pull-out effect of carbon fiber was dominant” in line 211 and 212 was corrected as “the pull-out effect of carbon fibers was dominant”.

“The AMGNS-MWCNT filler forms a good bridging effect” in line 213 and 214 was corrected as “The AMGNS-MWCNT fillers form a good bridging effect”

“carbon fiber” in line 214 was corrected as “carbon fibers”.

“carbon fiber” in line 216 was corrected as “carbon fibers”.

“and the carbon fiber extraction and the filler interaction” in line 216 was corrected as “and the extraction of carbon fibers and the interaction of fillers”.

“the AMGNS-MWCNT content” in line 225 was corrected as “the content of AMGNS-MWCNT”.

“without any filler” in line 227 was corrected as “without any fillers”.

“CNT is easy to be wound and agglomerated” in line 242 and 243 was corrected as “CNTs are easy to be wound and agglomerated”.

“its excellent performance” in line 243 was corrected as “their excellent performances”.

“GNS is also prone to pile-up in the matrix due to its van der Waals’ forces and π-π bonding” in line 243 and 244 was corrected as “GNS are also prone to pile-up in the matrix due to their van der Waals’ forces and π-π bonding”.

“its” in line 245 was corrected as “their”.

“The fibrous CNT effectively limits the agglomeration between GNS due to the π-π bond” in line 247 and 248 was corrected as “The fibrous CNTs effectively limit the agglomeration between GNS due to the π-π bond”.

“CNT” in line 248 was corrected as “CNTs”.

“the AMGNS-MWCNT content” in line 256 and 257 was corrected as “the content of AMGNS-MWCNT”.

Special thanks to you for your good comments. We appreciate for your warm work earnestly, and hope that the correction will meet with approval.
